# Learning Descriptive Image Captioning via Semipermeable Maximum Likelihood Estimation

**Zihao Yue, Anwen Hu, Liang Zhang, Qin Jin***
Renmin University of China
{yzihao, anwenhu, zhangliang00, qjin}@ruc.edu.cn
https://github.com/yuezih/SMILE

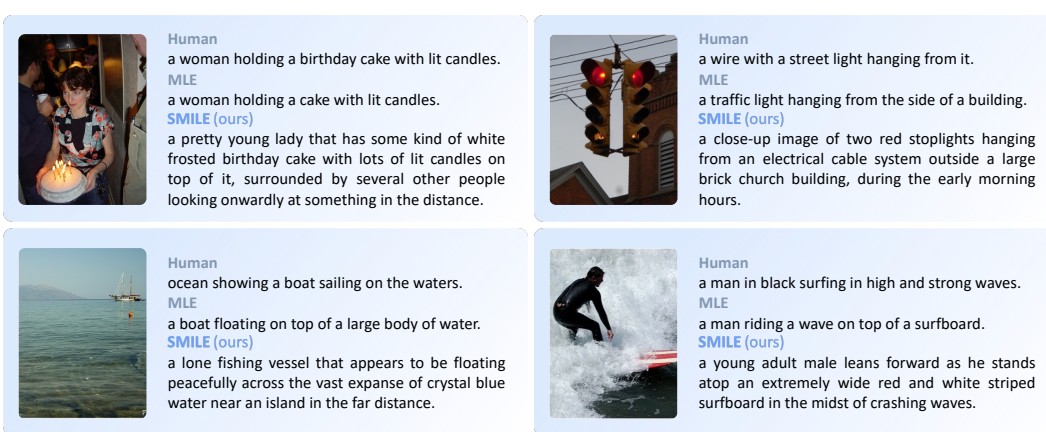

Figure 1: Descriptive captions generated by our SMILE-optimized captioning model, compared to human annotations and descriptions generated by the MLE-optimized captioning model.

## Abstract

Image captioning aims to describe visual content in natural language. As 'a picture is worth a thousand words', there could be various correct descriptions for an image. However, with maximum likelihood estimation as the training objective, the captioning model is penalized whenever its prediction mismatches with the label. For instance, when the model predicts a word expressing richer semantics than the label, it will be penalized and optimized to prefer more concise expressions, referred to as *conciseness optimization*. In contrast, predictions that are more concise than labels lead to *richness optimization*. Such conflicting optimization directions could eventually result in the model generating general descriptions. In this work, we introduce **S**emipermeable **M**ax**I**mum **L**ikelihood **E**stimation (SMILE), which allows richness optimization while blocking conciseness optimization, thus encouraging the model to generate longer captions with more details. Extensive experiments on two mainstream image captioning datasets MSCOCO and Flickr30K demonstrate that SMILE significantly enhances the descriptiveness of generated captions. We further provide in-depth investigations to facilitate a better understanding of how SMILE works.

---

*Corresponding Author.

37th Conference on Neural Information Processing Systems (NeurIPS 2023).

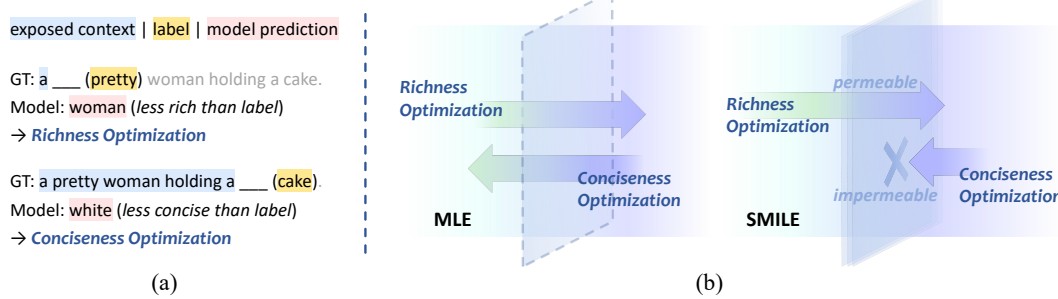

Figure 2: (a) Examples of richness optimization and conciseness optimization. When the label is "pretty" while the model predicts "woman" with less rich semantics, it leads to richness optimization; when the label is "cake" while the model predicts "white", conciseness optimization occurs. (b) Different from MLE, SMILE presents a 'semi-permeability' that accepts richness optimization while blocking conciseness optimization. Best viewed in color.

# 1   Introduction

The task of generating textual descriptions for a given image, commonly referred to as image captioning [47, 54], has long been plagued by the issue of overly-generic outputs [48, 49]. That is, models tend to generate similar descriptions for distinct images using simple concepts, but lack details (as shown in Fig. 1). Such generated captions cannot meet various needs, e.g., fine-grained online image search and recommendation, automatic data annotation for training vision-language models [4, 33], and poses a long-standing challenge in the field of image captioning. Considerable efforts have been devoted to generating more descriptive image captions, including constructing paragraph caption datasets with more details for training models [19], designing additional rewards [26, 28, 29], and prompting [27, 34] vision-language pre-training (VLP) models [59, 60] to enrich captions [55]. While these efforts yield some promising improvements, they seldom try addressing this problem by revisiting the core optimization objective.

Similar to text generation tasks such as machine translation [40], image captioning models are trained to predict the next word given context (the visual content and preceding words). The training is supervised by the maximum likelihood estimation (MLE) objective, which evaluates the model's predictive distribution over the vocabulary, and penalizes the model when it fails to predict the current label. However, unlike machine translation where the target sequence is relatively certain, describing an image can be quite diverse, and limited human annotations may not cover all accurate captions. Therefore, the strict supervision with MLE is not perfectly suitable for image captioning optimization. As shown in Fig. 2, when the model predicts a word (e.g., "woman") that is more 'concise' than the ground truth label (e.g., "pretty"), MLE computes a loss due to the mismatch and encourages the model to assign a higher probability to "pretty". From the model optimization perspective, this optimizes the model to possess the more descriptive captioning ability, a process referred to as *richness optimization* in this work. In contrast, if the model predicts a word (e.g., "white") expressing more details than the label (e.g., "cake"), MLE also penalizes the model and optimizes it towards a more concise captioning behavior (*conciseness optimization*). Since ground truth captions in commonly used datasets (e.g., MSCOCO [24]) are mostly simple (around 10 words), conciseness optimization is not rare during training, especially for finetuning VLP models [23, 60, 59]. But for image captioning, descriptions that contain more details rather than just generic concepts are usually preferred so that they can be used for accurately retrieving images or conveying more information for the visually impaired, etc. Therefore, conciseness optimization should be suppressed as much as possible, and richness optimization should be encouraged to incentivize the model to generate more descriptive captions.

In this work, to achieve this goal, we propose a simple but effective training objective for descriptive image captioning, namely Semipermeable MaxImum Likelihood Estimation (SMILE). Unlike MLE, which evaluates the predictive probability distribution over the entire vocabulary, SMILE considers the probability over a vocabulary subset with limited words, i.e., only containing **words in the ground truth caption**. Under typical conditions (with MLE), when a 'richer' word that conveys additional details gets a high confidence score (e.g., "white" in Fig. 2, and such details generally are not

included in the ground truth caption), it challenges the label word (e.g., "cake") in the competition of probability allocation. This competition consequently leads to a decrease in the probability assigned to "cake", thereby imposing a penalty on the model and thus resulting in conciseness optimization. However, with SMILE, probability allocation is limited to a subset of words that excludes "white". Regardless of the confidence score given to "white", it does not diminish the probability assigned to the label "cake", thus avoiding a loss increase. In this way, we suppress the conciseness optimization that would have been imposed with MLE. In contrast, when the label word expresses details (e.g., "pretty" in Fig. 2) while the model prefers a more 'concise' word "woman", it is highly likely that the more 'concise' word is part of the ground truth caption and therefore, included in the subset. Eventually, even if we only focus on the subset, the word still takes away the probability from the label word "pretty", resulting in model penalization. In this way, richness optimization is maintained. Our objective allows richness optimization but blocks conciseness optimization, similar to a semipermeable membrane. We, therefore, name this objective Semipermeable MLE (SMILE).

To verify the effectiveness of our proposed SMILE, we conduct extensive experiments on the image captioning task. The results demonstrate that SMILE can effectively optimize the model to generate significantly longer and more descriptive captions. Besides, we carry out abundant experiments to analyze how SMILE works, such as ablations on subset selection and descriptiveness origin, etc. Finally, we verify the generalization ability of SMILE on the video captioning task and discuss its limitation when facing other text generation tasks.

## 2 Method

### 2.1 Maximum Likelihood Estimation (MLE)

We first introduce the standard MLE for captioning optimization. As with many text generation tasks [40, 1], the model is trained to maximize the likelihood of the label when predicting the current word $w$ in the sequence given previous words $w_<$ and visual content $v$. The token-level loss function of MLE is defined as:

$$\mathcal{L}_{\text{MLE}} = -\sum_{j}^{|\mathcal{V}|} y_j \log \hat{P}^{\mathcal{V}}(w|w_<, v; \theta), \tag{1}$$

where the summation iterates over all words in the vocabulary $\mathcal{V}$, $y_j$ is the $j$-th element of the one-hot label vector, and $\hat{P}^{\mathcal{V}}$ is the predictive probability distribution over $\mathcal{V}$.

### 2.2 Semipermeable MaxImum Likelihood Estimation (SMILE)

Given a target sequence $D = [w_1, w_2, \ldots, w_N]$, we form a subset $\mathcal{V}_D$ that only includes the unique words occurred in $D$, defined as $\mathcal{V}_D = \{w_i \mid w_i \in D\}$. Then, the current word $w$ prediction is carried out within $\mathcal{V}_D$, with the loss function of SMILE defined as:

$$\mathcal{L}_{\text{SMILE}} = -\sum_{j}^{|\mathcal{V}_D|} y_j \log \hat{P}^{\mathcal{V}_D}(w|w_<, v; \theta). \tag{2}$$

Here, $\hat{P}^{\mathcal{V}_D}$ is the predictive distribution over the subset $\mathcal{V}_D$. Specifically, for the $j$-th word in $\mathcal{V}_D$, the probability $\hat{p}_j$ is calculated by the relative significance of the confidence score of the word compared to all other words in $\mathcal{V}_D$:

$$\hat{p}_j = \text{softmax}(\mathbf{z}_j) = \frac{\exp(\mathbf{z}_j)}{\sum_{k \in \mathcal{V}_D} \exp(\mathbf{z}_k)}. \tag{3}$$

The probability assigned to the label (the $j^*$-th word) determines the prediction loss. When a word $w^+$ expressing additional details beyond the ground truth caption gets a high confidence score (e.g., in Fig. 2, the label is "cake", $w^+$ is "white", and $\mathcal{V}_D = \{$"a", "pretty", "woman", "holding", "cake"$\}$, $w^+ \notin \mathcal{V}_D$), $w^+$ does not contribute to the denominator in Eq. (3) since it is not included in $\mathcal{V}_D$, and thus does not induce a decrease of the probability assigned to the label word $\hat{p}_{j^*}$, which would have otherwise increased the prediction loss. In other words, when the model tends to predict words

like $w^+$, these words do not participate in the probability allocation and do not take away any of the probability to be assigned to the label word, thus avoiding model penalization which leads to conciseness optimization. In contrast, when the model predicts a more 'concise' word $w^-$ (e.g., the label is "pretty" and $w^-$ is "woman"), such word $w^-$ usually is included in the ground truth caption, i.e., $w^- \in \mathcal{V}_D$. When allocating probability over $\mathcal{V}_D$, $w^-$ also participates and takes away some of the probabilities. Therefore, the model penalization is not avoided, and the induced richness optimization is maintained. Blocking conciseness optimization while allowing richness optimization is what SMILE is all about.

**SMILE is for further training.**   Since SMILE only considers the relative probability distribution within a subset, it does not penalize the model to generate detailed words beyond the subset, thus suppressing conciseness optimization. However, it makes sense only if the predicted detail is correct. Therefore, SMILE is applied to further train a model that has already been optimized with MLE, which ensures its fundamental captioning capability. Please note that SMILE is only adopted in the training phase. During inference, the model predicts across the entire vocabulary to determine the current generation.

**Initial context restriction**   It is well known that image captioning models are easy to suffer from exposure bias [2, 10, 36, 61]. When training in a teacher-forcing manner, the model is exposed to the ground truth context; however, during inference, each generation step is conditioned on the previous predictions. This causes a gap between training and inference. As SMILE does not require the model prediction to be consistent with the label as MLE does, the prediction is more prone to deviate from the label. Specifically, generating from the first token of the sequence, a prediction that is inconsistent with the label could lead to a completely unfamiliar context, exacerbating the exposure bias and affecting the autoregressive generation. Fortunately, we find that this issue can be effectively mitigated through some simple strategies, by ensuring that the initial context is correct, i.e., consistent with the label, which we name as *initial context restriction* and further discuss in Section 3.3.

## 3   Experiments

### 3.1   Experimental Setup

**Basic model and baselines**   Our proposed SMILE is architecture-agnostic and can be applied to any visual captioning model compatible with MLE optimization. As a representative, we validate SMILE on the base version of BLIP [23], one of the state-of-the-art vision and language models pre-trained with 129M images and paired captions. We first fine-tune BLIP on downstream datasets using MLE as the basic model, and then further optimize it with SMILE. For baselines, we compare our method with the latest descriptive image captioning solution, CapEnrich [55], and another two recent models NliCap [39] and GdisCap [49]. We also provide the performance of human-annotated ground truth captions for reference.

**Dataset**   We evaluate our method on the two most commonly used image captioning benchmarks, MSCOCO [24] and Flickr30K [57]. MSCOCO contains about 120K images, and we adopt the commonly used Karpathy splitting [18] with 5,000 images each for the validation and test sets. Flickr30K contains about 31K images, with 1,000 images each for the test and validation sets. For both datasets, each image has five human-annotated captions.

**Evaluation**   We evaluate the models from three aspects: descriptiveness, accuracy, and fluency. Descriptiveness refers to how detailed the caption describes the image. Following previous works [55], we evaluate descriptiveness by the performance of CLIP self-retrieval, which employs the CLIP model [33] as a retriever to recall the image with its caption from a candidates pool. This is based on the fact that captions with more details can better distinguish different images. The candidates pool is the hard retrieval pool constructed in the CapEnrich work [55], which additionally includes more similar images beyond the test set, placing a higher requirement on the descriptiveness of captions. We also report the average length of the generated captions and the lexical diversity (unique word count of all captions). Accuracy refers to the relevance of the generated caption to the image visual content, which we measure automatically by CLIPScore [13]. It calculates the semantic similarity between the image and the generated caption using a pre-trained CLIP model. For fluency, we report

Table 1: Image captioning performance of different methods on MSCOCO and Flickr30K. For human performance, we randomly select one annotation for each image for comparison with the others.

| Dataset | Method | Caption Length | Lexical Diveristy | Self-Retrieval R@1 | Self-Retrieval R@5 | CLIPScore | PPL |
|---|---|---|---|---|---|---|---|
| MSCOCO | NliCap | 9.5 | 0.8 | 2.9 | 9.3 | 75.5 | 67.8 |
| | GdisCap | 9.5 | 1.0 | 3.5 | 10.8 | 75.8 | 100.2 |
| | CapEnrich | 13.3 | 1.5 | 9.4 | 22.6 | **79.2** | 63.1 |
| | BLIP | 10.0 | 1.4 | 6.7 | 16.6 | 77.2 | 95.8 |
| | BLIP-$\mathcal{L}_{\text{SMILE}}$ | **22.3** | **4.5** | **10.0** | **24.5** | 75.0 | 95.6 |
| | Human | 10.4 | 4.1 | 7.6 | 20.0 | 77.6 | 129.1 |
| Flickr30K | CapEnrich | 15.2 | 1.0 | 29.2 | **54.9** | **81.5** | 67.1 |
| | BLIP | 11.6 | 0.8 | 25.4 | 46.4 | 78.8 | 65.6 |
| | BLIP-$\mathcal{L}_{\text{SMILE}}$ | **23.2** | **2.3** | **31.2** | 53.0 | 78.2 | 98.0 |
| | Human | 12.3 | 2.0 | 26.2 | 48.3 | 79.8 | 121.2 |

the language modeling perplexity[2] (PPL) of the captions with GPT-2 [32]. However, these automatic metrics have their limitations. For example, CLIP measures the overall semantics of the caption and struggles to focus on too many details. In addition, the evaluation models also suffer from the bias of the training data and thus may have difficulty in handling longer or more complex sentences. Therefore, we also conduct human evaluations, which will be discussed later.

## 3.2 Main Results

**Versus basic model** According to Table 1, compared with the basic BLIP model, SMILE significantly increases the average length of the captions (more than doubled on both datasets). SMILE leads to more detailed descriptions as well, greatly enhancing the model's self-retrieval performance. Although SMILE causes a slight decrease in the CLIPScore (-2.2 and -0.6 on MSCOCO and Flickr respectively), we consider this to be a reasonable phenomenon, as 'talks much errs much'. Besides, maintaining a comparable level of perplexity indicates that SMILE can incentivize longer and more detailed captions without compromising fluency.

**Versus baselines** As shown in Table 1, BLIP significantly outperforms all baselines on lexical diversity with SMILE optimization. Compared to traditional captioning methods that tend to generate dull texts with common words overused, SMILE effectively overcomes the lexical diversity bottleneck and outperforms humans. Our approach also achieves the best performance on self-retrieval. It is worth noting that the most competitive baseline CapEnrich requires automatically constructed data in a specific format, which integrates details from multiple manual captions of each image for training, along with carefully handcrafted or learnable prompts. In contrast, our approach is refreshingly simple, has fewer restrictions, and importantly, exhibits no incompatibility with other methods. Therefore, it can serve as an effective supplement or alternative to the currently employed solutions. Since our vanilla BLIP model with SMILE optimization already surpasses other methods that may require complicated design, we see no strong need for further attempts to combine SMILE with them in this work.

**Human evaluation** We randomly sample 100 images from the MSCOCO test set for human evaluation. For each image, we collect 4 candidate captions, including three captions generated by CapEnrich, basic model BLIP, and SMILE-optimized BLIP, respectively, and one human-annotated caption. All candidates are randomly shuffled. For each candidate caption, 5 human annotators are asked to independently rate on a scale of 1 to 5 (a higher score indicating better quality) from three aspects, namely descriptiveness, accuracy, and fluency. The 'descriptiveness' refers to the richness of the *correct* detail relevant to the image. Table 2 shows the average score of each model on the three aspects. SMILE substantially improves the descriptiveness of the captions, with a rate close to 5 (*excellent*), outperforming all other candidates including human-written captions by a large margin. A score beyond 4 for both accuracy and fluency demonstrates the good quality of captions

---

[2] https://huggingface.co/docs/transformers/perplexity

| Method | Semantic | | | Linguistic |
| --- | --- | --- | --- | --- |
| | Descrip. | Acc. | F1 | Flu. |
| CapEnrich | 3.89 | 4.39 | 4.12 | 4.36 |
| BLIP | 3.41 | 4.57 | 3.91 | **4.91** |
| BLIP-$\mathcal{L}_{\text{SMILE}}$ | **4.67** | 4.05 | **4.34** | 4.75 |
| Human | 3.53 | **4.53** | 3.97 | **4.87** |

Table 2: Human evaluation results of the captions for the randomly sampled 100 images. (score 1 to 5: *terrible*, *poor*, *fair*, *good*, and *excellent*)

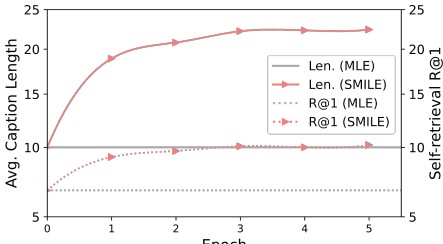

Figure 3: Trend of the average length and self-retrieval performance on the MSCOCO test set during training.

Table 3: Performance of different basic and further SMILE-optimized models. (*F.S.*: model trained **F**rom **S**cratch, i.e., BLIP with a randomly initialized text decoder trained on MSCOCO without pre-training; *PT*: pre-trained BLIP without fine-tuning; *PT+FT*: our default basic BLIP model with both pre-training and fine-tuning.)

| Row | Basic Model | Further Training | Caption Length | Lexical Diveristy | Self-Retrieval | | CLIPScore | PPL |
| --- | --- | --- | --- | --- | --- | --- | --- | --- |
| | | | | | R@1 | R@5 | | |
| 1 | *F.S.* | - | 9.6 | 0.6 | 1.9 | 6.3 | 73.9 | 56.2 |
| 2 | | $\mathcal{L}_{\text{SMILE}}$ | 15.4 | 0.7 | 0.7 | 2.5 | 63.7 | 86.7 |
| 3 | *PT* | - | 5.8 | 1.0 | 2.7 | 8.6 | 74.6 | 573.4 |
| 4 | | $\mathcal{L}_{\text{SMILE}}$ | 22.5 | 4.9 | 8.5 | 20.1 | 73.5 | 113.7 |
| 5 | *PT+FT* | - | 10.0 | 1.4 | 6.7 | 16.6 | **77.2** | 95.8 |
| 6 | | $\mathcal{L}_{\text{MLE}}$ | 10.0 | 1.4 | 6.5 | 16.6 | **77.2** | 67.6 |
| 7 | | $\mathcal{L}_{\text{SMILE}}$ | **22.3** | **4.5** | **10.0** | **24.5** | 75.0 | 95.6 |

by our SMILE-optimized captioning model, although it faces a higher risk of 'talks much errs much'. As descriptiveness emphasizes the recall of visual content while accuracy emphasizes description precision, we additionally calculate an F1 value of these two scores as the overall semantic score. As shown in Table 2, SMILE performs best when taking into account both aspects.

### 3.3 Ablation Study

**Training epochs** Fig. 3 presents the impact of training epochs on caption length and self-retrieval performance with SMILE optimization. It shows that SMILE takes only a few training epochs to achieve convergence and does not introduce too much extra training cost. For all of our models optimized with SMILE, we choose the checkpoints according to the best self-retrieval performance on the validation set, which always occurs within 3 epochs.

**Basic model** As the basic model used for SMILE optimization, BLIP is first pre-trained on a large-scale image-text dataset and then fine-tuned by MLE on downstream datasets. Both the pre-training (PT) and fine-tuning (FT) equip the model with fundamental image captioning capabilities. To investigate how such fundamental captioning ability influences SMILE's effectiveness, we compare the performance of different basic models and their further SMILE-optimized counterparts on MSCOCO. As shown in Table 3, for basic models, both pre-training and fine-tuning improve the model's performance in terms of descriptiveness and accuracy (rows 1, 3, and 5). After subsequent SMILE optimization, each basic model can generate much longer descriptions (rows 2, 4, and 7). Regarding how much SMILE improves descriptiveness, it is notable that when the fundamental captioning ability is poor, SMILE brings no improvement and can further decrease the performance (row 2 vs 1); but once the basic model is able to perform acceptable captioning, SMILE can contributes a lot (rows 4 vs 3 and 7 vs 5). This implies that SMILE requires the fundamental captioning ability of the basic model in order to further correctly generate more details. In addition, as SMILE optimization introduces extra training steps, we further train the default basic model with MLE and validate that simply adding training steps does not bring improvement (row 6 vs 5).

Table 4: Performance of SMILE-optimized models with different initial context restriction strategies.

| Initial Context Restriction | Caption Length | Lexical Diveristy | Self-Retrieval | | CLIPScore | PPL |
|---|---|---|---|---|---|---|
| | | | R@1 | R@5 | | |
| - | 17.0 | 3.0 | 6.8 | 16.2 | 71.9 | 82.0 |
| *shifting* | 21.3 | 4.0 | 9.6 | 22.4 | **75.0** | 85.2 |
| *default* | **22.3** | **4.5** | **10.0** | **24.5** | 75.0 | 95.6 |

Table 5: Performance comparison with different mixing ratios of MLE and SMILE on MSCOCO.

| $\lambda$ | Caption Length | Lexical Diveristy | Self-Retrieval | | CLIPScore | PPL |
|---|---|---|---|---|---|---|
| | | | R@1 | R@5 | | |
| 1 | 10.0 | 1.4 | 6.7 | 16.6 | **77.2** | 95.8 |
| 0.5 | 10.8 | 1.4 | 6.7 | 17.9 | 77.0 | 67.0 |
| 0.1 | 12.6 | 1.9 | 7.6 | 18.2 | 76.1 | 69.3 |
| 0.05 | 14.7 | 2.3 | 8.7 | 20.9 | 76.4 | 74.1 |
| 0.01 | 19.8 | 3.6 | **10.9** | **25.1** | 76.2 | 79.4 |
| 0 | **22.3** | **4.5** | 10.0 | 24.5 | 75.0 | 95.6 |

**Initial context restriction**  As aforementioned in Section 2.2, we propose initial context restriction to alleviate exposure bias for SMILE optimization. We design two implementations, *First-token MLE* and *First-token Shifting*, to achieve such restriction:

1. *First-token MLE* adopts MLE loss for the first token of the target sequence, and SMILE for the remaining tokens.

2. *First-token Shifting* shifts the label of the first token of the target sequence to its alternative. For example, in the vocabulary of the BERT model [7], the word "a" can be replaced with its subword form "##a", which rarely appears to be the first token in the training corpus. By doing so, we replace the familiar word with an unfamiliar one, making it difficult for the model to predict correctly (even within a subset). This often results in model penalization, making the prediction of the first token more likely to be consistent with the ground truth.

Since the shifting approach is designed with certain requirements on the model vocabulary, making it potentially difficult to be applied to all models, we use the *First-token MLE* by default to address the exposure bias issue. To demonstrate the effectiveness of this strategy, in Table 4, we show that without initial context restriction, SMILE optimization can not achieve promising results. Either the default *First-token MLE* implementation or the *First-token Shifting* implementation helps alleviate exposure bias and the former is better.

**Mixing $\mathcal{L}_{\text{MLE}}$ with $\mathcal{L}_{\text{SMILE}}$**  Since SMILE improves descriptiveness at the expense of some accuracy, it could pose a risk in scenarios that demand higher accuracy. To strike a balance, we carry out experiments where the overall learning objective combines both $\mathcal{L}_{\text{MLE}}$ and $\mathcal{L}_{\text{SMILE}}$, defined as:

$$\mathcal{L}_{\text{overall}} = \lambda \cdot \mathcal{L}_{\text{MLE}} + (1 - \lambda) \cdot \mathcal{L}_{\text{SMILE}}, \lambda \in [0, 1]. \tag{4}$$

Table 5 illustrates how the balance between MLE and SMILE impacts the performance in terms of descriptiveness and accuracy. As $\mathcal{L}_{\text{SMILE}}$ becomes more dominant, the model tends to generate longer outputs with higher lexical diversity and descriptiveness; however, accuracy correspondingly decreases. This demonstrates that a compromise between descriptiveness and accuracy can be achieved by simply combining $\mathcal{L}_{\text{SMILE}}$ with $\mathcal{L}_{\text{MLE}}$. Notably, we find that when $\mathcal{L}_{\text{MLE}}$ is incorporated at a very low ratio (0.01), the generated captions achieve the best performance in self-retrieval. This suggests that while SMILE facilitates more details, ensuring accuracy is also important for captions to better describe and distinguish images.

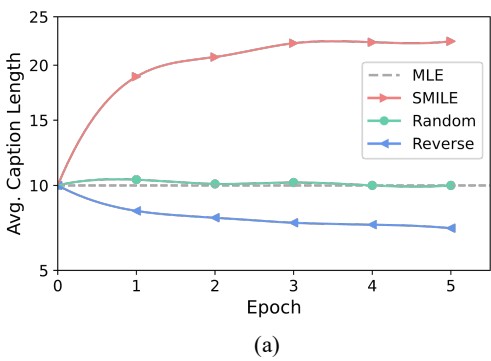 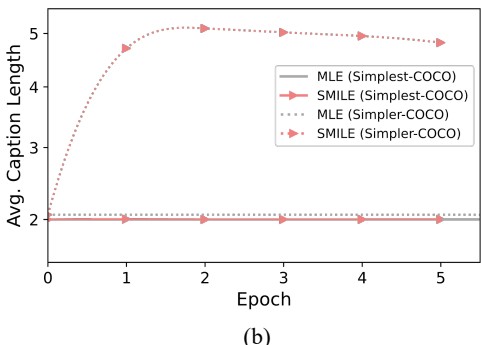

|  (a) | (b) |

Figure 4: Average caption length during training with (a) different subsetting strategies and (b) on different training corpus.

## 3.4 Further Analysis

In this section, we conduct experiments to further analyze how SMILE works. As we hypothesized, SMILE implements a 'semi-permeability' to retain only richness optimization, leading the model to generate more details. Therefore, we **first** investigate whether such an effect is attributed to our subset selection. **Then**, we affirm the 'semi-permeability' by flipping it to observe an opposite property of SMILE. **Besides**, since BLIP-$\mathcal{L}_{\text{SMILE}}$ surpasses human annotation in descriptiveness as shown in Table 1 and Table 2, we analyze where it gains the capability to generate details. **Furthermore**, we provide a visualization to illustrate how SMILE exerts its effect from the token-level loss. **Finally**, we test the generalization ability of SMILE on video captioning.

**Subset selection is key in SMILE.** We argue that the effectiveness of SMILE is attributed to our subset selection strategy that selects the target sequence words. To verify this, we ablate the SMILE subsetting strategy by replacing it with *Random subsetting*, that is, for a given target sequence, we randomly choose 10 words (the average caption length of MSCOCO) from the entire vocabulary to form the subset, ensuring the subset size is relatively consistent with that of SMILE. Fig. 4 (a) illustrates the impact of SMILE and Random subsetting on the generation length during training. Random subsetting maintains the describing habits of the basic model, neither lengthening nor shortening the generation length. This is due to that the model's preferred predictions among the entire vocabulary, whether more or less 'rich' than the label, are less likely to be included in the randomly-chosen subset. This leads to a mechanism akin to 'bidirectional impermeability', where both conciseness and richness optimization are weakened. This also demonstrates that it is the subset selection of SMILE that contributes to the increasing length of model generation.

**Reverse subsetting flips the 'semi-permeability'.** Since SMILE achieves a 'semi-permeability' which allows richness optimization while blocks conciseness optimization, it is natural to assume that such 'semi-permeability' could be flipped with a reversed subset selection. Thus, we conduct experiments by proposing *Reverse subsetting*, that is, given a target sequence $D = [w_1, w_2, \ldots, w_N]$, when predicting each word, we select the complement of the target sequence word set in the entire vocabulary, plus the current label, as the subset. Hence, for the prediction of the $i$-th word, the corresponding subset is $\mathcal{V}_{S_i} = \{w | w \notin \mathcal{V}_D\} \cup \{w_i\}$. As shown in Fig. 4 (a), with a subset complementary to that of SMILE, Reverse subsetting exhibits the opposite property of SMILE, leading the model to generate increasingly shorter captions. With these observations, we confirm that there is a 'semi-permeability', which is achieved by our subsetting strategy and can be inverted.

**Models 'absorb' descriptiveness from the corpus.** Although SMILE achieves unidirectional richness optimization, one may still wonder where models with SMILE optimization learn the descriptive captioning ability to surpass human annotators. We suspect that models could learn the usage of details from the whole corpus. To verify this hypothesis, we construct two datasets based on MSCOCO, namely Simplest-COCO, which contains no details, and Simpler-COCO, with only a few details. Concretely, for Simplest-COCO, we extract the subject from the caption and prefix it with a definite article based on grammatical rules, such as "a girl". For Simpler-COCO, we extract the subject-verb-object constituent of the caption as a new caption with few details, such as "a man working". All the mentioned natural language processing (e.g., parsing and part-of-speech tagging)

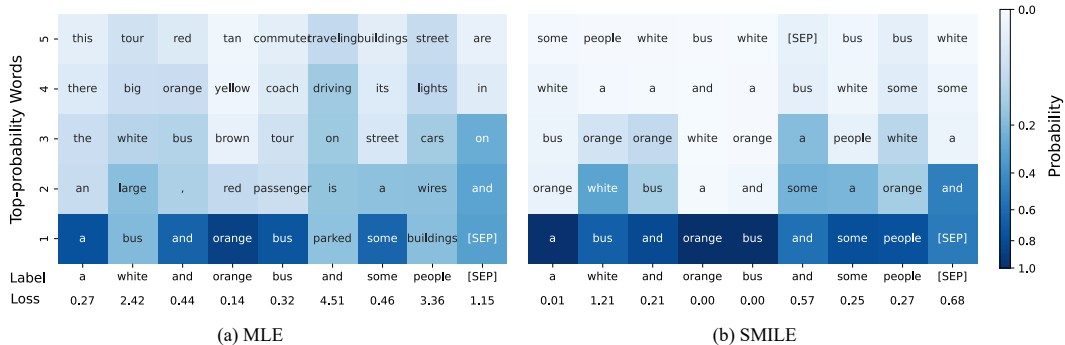

Figure 5: Token-level model predictive distribution and penalization of (a) MLE and (b) SMILE. The model is less penalized if the label word is assigned a higher probability.

Table 6: Video captioning performance on MSR-VTT.

| Method | Caption Length | Lexical Diveristy | EMScore | | | Self-Retrieval | | PPL |
|---|---|---|---|---|---|---|---|---|
| | | | P | R | F | R@1 | R@5 | |
| BLIP4video | 8.1 | 1.1 | 26.4 | 28.1 | 27.2 | 19.0 | 38.7 | 150.1 |
| BLIP4video-$\mathcal{L}_{\text{SMILE}}$ | **20.6** | **4.5** | **26.6** | **28.6** | **27.5** | **27.3** | **50.0** | 163.0 |
| Human | 9.2 | 3.8 | **26.6** | 28.2 | 27.4 | 23.0 | 42.8 | 561.8 |

is performed with SpaCy [16]. The average caption length for Simplest-COCO and Simpler-COCO is 2.0 and 2.5, respectively. On these two datasets, we first train the default basic model with MLE until convergence and further optimize it with SMILE. As shown in Fig. 4 (b), only MLE optimization does not achieve longer descriptions than the training corpus on both datasets. On the other hand, with SMILE optimization, the description length does not increase when training on Simplest-COCO; however, on Simpler-COCO, the model generates captions about twice as long as the ground truths. This suggests that richness optimization relies on the details in the whole training corpus, and SMILE helps models 'absorb' the capability to express more details.

**Visualization of token-level penalization, MLE versus SMILE**   Taking a sample from the MSCOCO training set as an example, Fig. 5 shows the predictive distribution and loss value for each word during MLE and SMILE optimization, respectively. Note that the predictions in Fig. 5 (a) and (b) are given by an identical model but the former distribution is calculated over the whole vocabulary, while the latter over the SMILE subset. When predicting the second word ("white" in the target sequence), the model prefers a more 'concise' word "bus" than the label "white". For MLE, "bus" takes away much of the probability, resulting in a significant loss that causes richness optimization. Similar to MLE, SMILE also penalizes such prediction since both "white" and "bus" are included in the subset. When predicting the sixth word (the label word is "and"), the model prefers a more 'detailed' word "parked" to additionally describe the state of the bus. Due to such a mismatch, MLE still imposes a heavy penalty with a loss of 4.51; however, SMILE performs a more lenient optimization with a loss of 0.57. This is because the word "parked" is not contained in the subset, and by allocating probabilities over the subset, the label word "and" obtains the highest predictive probability. Therefore, SMILE circumvents the penalty induced by a more 'detailed' prediction and blocks conciseness optimization.

**SMILE generalizes well to video captioning.**   We conduct experiments on MSR-VTT [53] to test the generalization ability of SMILE on the video captioning task. MSR-VTT consists of 10K videos and each video clip is annotated with 20 English sentences. The BLIP4video [58], a modified version of BLIP designed for video tasks, is chosen as our basic model. As for evaluation, we apply the reference-free metric EMScore [38]. It evaluates embedding matching between video and caption in terms of precision (P), recall (R), and the F1 score (F), to measure the accuracy and descriptiveness of the captions. We also provide the self-retrieval performance, with the F1 score of EMScore for retrieving, and the test set as the candidates pool. Table 6 shows that SMILE also enables the model to generate longer descriptions on video captioning, and brings an overall improvement in both accuracy and descriptiveness.

# 4 Related Works

In this work, we focus on generating descriptive image captions from the perspective of the language modeling objective. Existing works also propose different strategies to increase the descriptiveness of captions. For example, Gu et al. [11] and Liu et al. [26] design two-stage decoding strategies with caption generation and refinement. Liu et al. [26] and Shi et al. [39] integrate descriptive details from multiple-sentence annotations with additional rewards or natural language inference (NLI) relations. In addition to descriptiveness, existing works also employ contrastive learning [3, 6, 29] and self-retrieval strategy [28, 45] to increase the discriminability of captions. Wang et al. [48] re-weight the ground truth captions to emphasize more discriminative ones. Wang et al. [49] highlights the uniqueness of each image within groups with similar images. More recently, CapEnrich [55] stimulates VLP models [59, 60] to express more details with prompting strategies [27, 34]. It achieves state-of-the-art performance on the descriptiveness and discriminability of image captioning. However, CapEnrich requires training on automatically constructed data with additional utterances, which integrate details from multiple captions annotated for each image. This leads to unconventional text structure and thus affects the fluency of the generated caption. In contrast to the above-mentioned methods, SMILE does not rely on either multiple annotated captions or the construction of additional data. To the best of our knowledge, SMILE is the first approach to enhance models' descriptive captioning capability by focusing on the language modeling objective itself.

Besides, text generation models trained with MLE tend to generate dull texts with high-frequency words [52, 37]. Attempts for fixing such degeneration issue including variant decoding methods [22, 46, 20, 14, 41] and learning algorithms [12, 9, 15, 51]. In this work, SMILE-optimized models output with much greater lexical diversity, also demonstrating the capacity of mitigation of such problems besides improving descriptiveness.

# 5 Conclusion

For descriptive image captioning, we revisit the commonly used maximum likelihood estimation (MLE) training objective. We argue that MLE is not perfectly suitable for image captioning tasks because of two conflicting optimizations: conciseness optimization and richness optimization. To steer the model toward generating more descriptive captions, we propose to mitigate conciseness optimization and maintain richness optimization by revising the objective with a simple but effective vocabulary subset selection strategy, namely Semipermeable MaxImum Likelihood Estimation (SMILE). Extensive experiments validate that SMILE helps models to generate much longer and more descriptive captions without complex architecture design or extra data annotation.

# 6 Limitations

Technically, it is possible to apply SMILE to any autoregressive text generation task with an MLE objective. To investigate this, we conduct a preliminary exploration on the Supervised Fine-tuning [31] (SFT) of Large Language Models [44, 43] (LLMs) – fine-tuning LLMs to follow human instructions, such as writing and answering questions. Unfortunately, comparing SMILE with MLE for SFT, we don't observe significant improvement in response length or overall quality (see Appendix B). One potential factor is that the target response in the training corpus is often very long (for instance, a paragraph exceeding 100 words). This causes that the constructed subsets are significantly larger than the ones in the image captioning task, and differ less from the entire vocabulary set. Therefore, SMILE may struggle to be directly applied to other text generation tasks. Further exploration of this issue will be reserved for future work.

## Acknowledgements

This work was partially supported by the Beijing Natural Science Foundation (L233008), the National Natural Science Foundation of China (No. 62072462) and the National Key R&D Program of China (No.2020AAA0108600).

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

# A SMILE Implementation

We present pseudocode in Algorithm 1 to better understand SMILE, which can be implemented and applied in such a straightforward manner. Further details including *Reverse Subseting* and *Random Subseting* can be found in our released code at `https://github.com/yuezih/SMILE`.

---

**Algorithm 1** Pseudocode of SMILE in a PyTorch-like style.

---

```
# B: batch size
# N: sequence length
# V: vocabulary size
# logits: language model prediction logits (B x N x V)
# labels: ground truth label at each position (B x N)

if objective == 'MLE':
    loss = CrossEntropyLoss(logits, labels)

elif objective == 'SMILE':
    # each sequence gets a unique mask to mask words not in the sequence
    logits_mask = zeros(B, V).scatter_(1, labels, True) # B x V

    # expand for every position in the sequence
    logits_mask = logits_mask.unsqueeze(1).expand(-1, N, -1).clone() # B x N x V

    # perform first-token MLE (optional)
    logits_mask[:, 0, :] = True # B x N x V

    # apply mask to the logits
    selected_logits = logits.masked_fill(logits_mask==False, float('-inf')) # B x N x V

    loss = CrossEntropyLoss(selected_logits, labels)
```

---

scatter_: scatter values into a matrix by indices (`https://pytorch.org/docs/stable/generated/torch.Tensor.scatter_.html`); CrossEntropyLoss integrates the softmax operation.

Table 7: Performance of LLaMA-13B fine-tuned using MLE and SMILE, respectively, on the Vicuna evaluation framework. Scores are relative values re-scaled by taking the scores of reference answers as full marks (100). Eps.: Training epochs; Obj.: Learning objective; Len.: Average answer length; Lex.: Lexical diversity.

| Eps. | Obj. | Len. | Lex. | Gene. | Know. | Role. | Com.Sen. | Fermi | Count. | Writ. | Overall |
|------|------|------|------|-------|-------|-------|----------|-------|--------|-------|---------|
| 5    | $\mathcal{L}_{\text{MLE}}$ | 94.8 | 2.4 | 82.3 | **88.2** | 82.9 | 85.6 | 77.4 | 85.3 | 83.1 | 83.5 |
|      | $\mathcal{L}_{\text{SMILE}}$ | **95.8** | 2.4 | **92.4** | 86.2 | **85.3** | **90.4** | **79.6** | **85.4** | **85.8** | **86.4** |
| 10   | $\mathcal{L}_{\text{MLE}}$ | 89.2 | 2.3 | 92.9 | 88.9 | **85.8** | 86.9 | 78.3 | 84.4 | **84.7** | **86.0** |
|      | $\mathcal{L}_{\text{SMILE}}$ | **96.1** | 3.4 | **97.3** | **89.0** | 82.4 | **87.8** | **79.2** | **87.4** | 75.6 | 85.5 |
| 20   | $\mathcal{L}_{\text{MLE}}$ | 90.2 | 2.3 | 92.0 | 88.0 | 88.2 | **91.3** | 75.7 | **86.5** | **82.2** | **86.3** |
|      | $\mathcal{L}_{\text{SMILE}}$ | **96.4** | 3.5 | **93.4** | **89.2** | **91.1** | 90.6 | **81.6** | 85.3 | 62.2 | 84.8 |

# B SFT of LLMs

Supervised fine-tuning (SFT) aims to enhance the capacity of large language models (LLMs) to follow human instructions better [31]. This section reports our experimental findings comparing SFT tasks carried out with MLE versus SMILE. We choose the widely used LLaMA-13B [44] as the basic model, and fine-tune it on the Alpaca dataset [42], which contains 52,000 instructions and demonstrations generated by the text-davinci-003 engine[3]. To ensure fair comparisons, the pre-trained basic model is fine-tuned using both MLE and SMILE, maintaining identical settings for both objectives. Each experiment involves fine-tuning the baseline model using a Low-Rank Adaptation (LoRA) approach [17] for parameter efficiency, employing 4 RTX A6000 nodes with a batch size of 8. The fine-tuning process is conducted using the LMFlow framework [8]. For evaluation, we employ

---

[3]`https://platform.openai.com/docs/models/gpt-3-5`

Table 8: Performance of different basic models before and after SMILE optimization.

| Model | $\mathcal{L}_{\text{SMILE}}$ | Caption Length | Lexical Diversity | Self-Retrieval | | CLIPScore | PPL |
|---|---|---|---|---|---|---|---|
| | | | | R@1 | R@5 | | |
| BLIP_base [23] | | 10.0 | 1.4 | 6.7 | 16.6 | **77.2** | 95.8 |
| | ✓ | **22.3** | **4.5** | **10.0** | **24.5** | 75.0 | 95.6 |
| BLIP_large [23] | | 10.0 | 1.4 | 6.7 | 17.3 | **77.4** | 92.3 |
| | ✓ | **24.0** | **4.9** | **9.2** | **22.4** | 74.7 | 100.0 |
| OFA_base [50] | | 10.1 | 1.2 | 5.9 | 15.6 | **76.7** | 64.6 |
| | ✓ | **16.5** | **4.0** | **6.9** | **16.2** | 71.6 | 109.7 |
| OFA_large [50] | | 9.8 | 1.2 | 6.4 | 16.0 | **77.1** | 68.6 |
| | ✓ | **18.8** | **5.9** | **9.3** | **21.9** | 73.6 | 125.9 |
| mPLUG [21] | | 9.7 | 1.3 | **5.7** | **15.5** | **76.6** | 59.6 |
| | ✓ | **12.3** | **2.8** | 3.9 | 11.0 | 71.0 | 92.3 |
| VinVL [60] | | 9.9 | 1.3 | **5.5** | **14.4** | **76.9** | 60.6 |
| | ✓ | **25.2** | **5.6** | 5.4 | 13.8 | 71.9 | 113.9 |

the evaluation framework proposed in Vicuna [5]. It contains eight question categories, including generic (Gene.), knowledge (Know.), roleplay (Role.), common-sense (Com.Sen.), Fermi problems (Fermi), counterfactual (Count.), coding/math tasks, and writing (Writ.), with ten questions for each category. The model's answer is rated by GPT-4 [30] in terms of helpfulness, relevance, accuracy, and detail, by an overall score ranging from 1 to 10. To ensure consistent independent scoring, GPT-4 rates the candidate answer by comparing it with a reference answer (from GPT-3.5-turbo[3]). Given that GPT-4 is not an efficient judge for coding/math tasks [5], and LLaMA-13B also struggles with these tasks, we exclude coding/math tasks in the evaluation.

Table 7 shows the performance of the models after fine-tuned for 5, 10, and 20 epochs using both MLE and SMILE. Compared to MLE, SMILE enables the model to generate slightly longer responses with enhanced lexical diversity. In terms of answer quality, the SMILE-optimized model demonstrates advantages across almost all task types at the early stages of training (after 5 epochs). However, these advantages diminish as the training process continues, particularly in the writing task, where the average quality of answers from the SMILE-optimized model notably declines. In conclusion, the significant effectiveness of SMILE in the SFT task is not evident. As stated in the Limitations section (Section 6) in the main paper, SMILE faces challenges in extending to other text generation tasks.

## C   Cross-model Generalizability

Since the aforementioned experiments validate the efficacy of SMILE on the base version of BLIP (and BLIP4video), it is crucial to investigate the generalizability of SMILE across different model architectures and sizes. Thus, we apply SMILE to several popular VLP models across scales [23, 50, 21, 60]. The implementation details are consistent with their respective officially released codes. Table 8 displays the performance of these models on MSCOCO. For all models, SMILE notably increases the average length and lexical diversity of the captions, indicating its 'semi-permeability' can generalize well to different model architectures and scales. Besides, for BLIP and OFA, SMILE also significantly enhances their descriptiveness, as evidenced by improved self-retrieval performance. However, such gains are not observed in mPLUG and VinVL. While the generated captions become longer with SMILE optimization, their accuracy is adversely affected, leading to dropped self-retrieval performance and CLIPScore. Regarding this discrepancy on different models, one possible reason might be that mPLUG and VinVL adopt Prefix Language Modeling (PrefixLM) [35] or Masked Language Modeling (MLM) [7] styles pre-training for text generation, whereas BLIP and OFA utilize Causal language modeling (CLM). We hypothesize that CLM is intrinsically more advantageous for text generation tasks, potentially enabling models to develop a generation pattern that is more robust and adaptable in embracing SMILE optimization. We leave further exploration on this matter for future work.

Table 9: Comparison of SMILE-optimized models and MLLMs.

| Model | Caption Length | Lexical Diversity | Self-Retrieval | | CLIPScore | PPL |
| --- | --- | --- | --- | --- | --- | --- |
| | | | R@1 | R@5 | | |
| BLIP-$\mathcal{L}_{\text{SMILE}}$ | 22.3 | 4.5 | 10.0 | 24.5 | 75.0 | 95.6 |
| BLIP-$\mathcal{L}_{\text{MLE+SMILE}}$ | 19.8 | 3.6 | **10.9** | **25.1** | 76.2 | 79.4 |
| MiniGPT-4 | **66.3** | **6.4** | 10.5 | 23.8 | 75.6 | 14.8 |
| LLaVA | 32.5 | 3.8 | 9.7 | 23.4 | **79.7** | 20.0 |

Table 10: Performance of models trained on MSCOCO with and without paraphrases.

| Method | Paraphrase | Training Data | Caption Length | Lexical Diversity | Self-Retrieval | | CLIPScore | PPL |
| --- | --- | --- | --- | --- | --- | --- | --- | --- |
| | | | | | R@1 | R@5 | | |
| BLIP | ✓ | 567K | 10.0 | 1.4 | 6.7 | 16.6 | 77.2 | 95.8 |
| BLIP-$\mathcal{L}_{\text{SMILE}}$ | ✓ | 567K | 22.3 | 4.5 | 10.0 | 24.5 | 75.0 | 95.6 |
| BLIP-$\mathcal{L}_{\text{SMILE}}$ | | 113K | 17.9 | 3.3 | 8.9 | 21.3 | 75.2 | 90.3 |

# D   Comparison with Multimodal LLMs

With the advent of the LLM era, various Multimodal LLMs (MLLMs) [56] are developed to facilitate general multimodal tasks, including descriptive image captioning. Therefore, we conduct a shallow evaluation with two popular MLLMs, MiniGPT-4 (7B) [62] and LLaVA (Lightning-MPT-7B) [25], to compare them with our SMILE-optimized models. Models are prompted to generate captions with an example prompt from the LLaVA project, namely `Describe the following image.`. As shown in Table 9, BLIP with SMILE achieves comparable self-retrieval performance with MLLMs, while underperforming them obviously in both caption length (MiniGPT-4) and accuracy (LLaVA). However, it is important to note that such a direct comparison might be somewhat unfair. While MLLMs employ additional training data (e.g., detailed description) to learn to describe in detail, SMILE enables models to directly learn from existing data (e.g., short captions in MSCOCO) for more descriptive captioning. Besides, our model size ($\sim$227M) and pre-training data scale are significantly smaller than MLLMs. Despite the strong capabilities demonstrated by LLMs, the heavy computational resources they rely on are worth noting.

# E   Paraphrases Ablation

Captioning datasets including MSCOCO and Flickr30K usually contain multiple annotations for one image or video [24, 57, 53], a.k.a., paraphrases. This may raise questions about whether the gains in descriptiveness come from combining these paraphrases, rather than 'absorbing' from the entire training dataset. On this topic, we test with a no-paraphrase corpus, keeping just one caption per image for training in MSCOCO. As shown in Table 10, by learning from a corpus without paraphrases, the model can still boost descriptiveness, with remarkable improvement in terms of caption length, lexical diversity, and self-retrieval performance. This suggests that paraphrases in training data are not the key to descriptiveness gains.

