# OpenReview forum: "Learning Descriptive Image Captioning via Semipermeable Maximum Likelihood Estimation"
_NeurIPS.cc/2023/Conference — NeurIPS 2023 poster_

### Official Review · Reviewer_nk4n · 2023-06-18

**Soundness:** 2 fair
**Presentation:** 3 good
**Contribution:** 2 fair
**Rating:** 6
**Confidence:** 5

**Summary:**

## Summary

- In the paper, the authors introduce a new method for generating descriptive captions for images. They call it SMILE, which stands for "Semipermeable Maximum Likelihood Estimation." Unlike traditional image captioning models that often produce generic descriptions, SMILE allows for more detailed and rich descriptions while still maintaining accuracy.
- Image captioning differs from Machine Translation in that there are numerous ways to describe an image, and limited human captions cannot encompass the entirety of possible descriptions. To address this issue, SMILE utilizes information from the entire corpus. The model used is BLIP, which is first fine-tuned with standard MLE loss and then further fine-tuned with SMILE loss. The dataset used for this project includes MSCOCO and Flickr30K.
- SMILE only operates at training time.
- The key idea in SMILE is restricting the vocabulary to only the unique words present in the target caption during decoding.
- Since most generation models are trained with teacher forcing, they might suffer from exposure bias. Since SMILE does not require prediction to be consistent with ground truth labels, it is more prone to deviate from the ground truth label. To alleviate this problem, the authors propose two techniques
	- First Token MLE
		- MLE loss for first token, SMILE loss for remaining tokens in the sequence.
	- First Token Shifting
		- Change the label of first token to a similar token "a" -> "##a" where ## denotes the subword prefixed with a space. This results in model making a mistake on first token and encourages the model to get the first token right during the SMILE training and subsequently during inference.
- The metrics used are CLIP score, caption length, lexical diversity, self-retrieval@1 and @5, and perplexity for measuring fluency.
- The baseline models compared against are NliCap, GdiCap and CapEnrich which are quite standard for descriptive image captioning. The authors also do human evaluation on a small subset (50 examples).
- Results indicate that the SMILE technique is able to generate more descriptive captions.


####  nit
- Typo Line 259 SMLIE -> SMILE

**Strengths:**

- Paper tackles a significant problem in image captioning.
- Easy to read, prior work is cited well, and results are promising.

**Weaknesses:**

- Most metrics used are automatic, and the human evaluation set is very small.
- Paper could also benefit from additional details around human evaluation experiments.

**Questions:**

### Remarks and Questions

- All metrics used in the paper are automatic metrics. Have you considered using RefCLIPScore instead of CLIPScore, which also gives some weight to the reference captions?
	- Perhaps another metric to consider could be running an object detector on the image (like RCNN and collecting all objects correctly identified in the caption)
- Can you provide more details on human evaluation? 50 seems a very small number of samples to justify the results conclusively.  In Table 2, descriptiveness improves to 4.64, but there is also a drop in accuracy compared to the last row.
- Both MSCOCO And Flickr30K datasets provide multiple captions. It is unclear how the multiple reference captions are used from reading the paper. While constructing the subset vocabulary, do we take a union of all tokens from all the captions for an image?
	- Maybe the gains in descriptiveness are coming from combining all captions and the analysis presented in the section "Models absorb details from the corpus."
- For the video captioning task, the human perplexity is quite high, 561.8, which is significantly higher than other numbers, any idea why this is the case?

**Limitations:**

The authors have appropriately addressed the limitations.

---

> ### Author Rebuttal · Authors · 2023-08-09
>
> Thank you for the insightful understanding and valuable feedback. We address each of your concerns and questions as follows:
>
> > **Q1: (a)** All metrics are automatic metrics. Have you considered RefCLIPScore instead of CLIPScore? **(b)** Perhaps another metric could be running an object detector.
>
> **A1 (a):** As shown in Fig. 1, SMILE encourages much richer captions than the ground truth, and such **mismatches** could lead to **unfair penalization** by reference-based metrics, including RefCLIPScore. Thus, we did not consider these metrics. We concur with your concerns regarding automatic metrics, which prompt us to conduct additional human evaluations.
>
> **A1 (b):** Thank you for proposing detection-based metrics. We recognize that the details in captions might be multifaceted, e.g., object attributes and states, and it is difficult for object detection methods to fully reflect these details. However, we agree that object information is an important aspect of descriptiveness, so we added a corresponding evaluation with **UMIC** [1], where the image encoder takes the Faster R-CNN features as input, enabling a better grasp of objects in images. The table below shows UMIC evaluation on MSCOCO, where SMILE notably boosts the basic model's performance. It's worth noting that the best-performing method, CapEnrich, introduces auto-constructed training data by extracting scene graphs from multiple reference captions, which could benefit object recall.
>
> Method|UMIC
> -|:-:
> NliCap|65.5
> GdisCap|67.4
> CapEnrich|**71.5**
> BLIP|68.2
> BLIP-SMILE|69.9
> Human|71.1
>
> [1] UMIC: An Unreferenced Metric for Image Captioning via Contrastive Learning, ACL 2021.
>
> > **Q2: (a)** Can you provide more details on human evaluation? **(b)** 50 seems to be a very small number of samples. **(c)** In Table 2, descriptiveness improves but accuracy drops.
>
> **A2 (a):** Our human evaluation was a blind review process, with each caption candidate scored by five annotators on three dimensions: descriptiveness, accuracy, and fluency. A score of 5 indicates "*excellent*", e.g., no errors or inaccuracies (in accuracy), or including rich details (in descriptiveness). Conversely, a score of 1 suggests that the caption is "*terrible*", e.g., entirely wrong or describes minimal visual content. To help better understand the human evaluation results, we've provided some real examples in ***Global Response***. We will enhance the description of human evaluations in our revision.
>
> **A2 (b):** We agree that 50 cases may not sufficiently justify our results. Thus, we've conducted a new round of human evaluation with **another 50 random cases**. While 100 cases is still a modest sample set, it entails a substantial workload of 1200 ratings for each annotator. The updated results (100 cases) shown in the following table align with our initial findings.
>
> Method|Descrip.|Acc.|F1|Flu.
> -|:-:|:-:|:-:|:-:
> CapEnrich|3.89|4.39|4.12|4.36
> BLIP|3.41|**4.57**|3.91|**4.91**
> BLIP-SMILE|**4.67**|4.05|**4.34**|4.75
> Human|3.53|4.53|3.97|4.87
>
> **A2 (c):** As highlighted in our paper, "**Talks much errs much.**" Generic captions (~10 words long) often contain simple concepts to describe the main features in images, and thus are less prone to errors. In contrast, SMILE captions (over 20 words on average) contains much more details, amplifying the risk of errors, especially when these details are inherently harder to identify (e.g., small objects). We noticed that the annotators often base accuracy assessment on **error spotting**, meaning simple captions easily get perfect scores (i.e., 5). However, for very detailed captions, even a slight inaccuracy can reduce scores (to 4 or less), greatly disadvantaging SMILE in human evaluations. We've illustrated this with examples in ***Global Response***. Nevertheless, it's worth noting that SMILE captions still scored above 4 on average.
>
> Finally, if accuracy is still a concern, our *Supplementary* experiments show that descriptiveness and accuracy can be **balanced** by combining SMILE with MLE, and such a trade-off can lead to enhanced accuracy and self-retrieval performance.
>
> > **Q3: (a)** It is unclear how the multiple reference captions are used. **(b)** Maybe the gains in descriptiveness are coming from combining all captions and the paper analysis.
>
> **A3 (a):** Thank you for your in-depth understanding. We didn't have any specific scheme to leverage multiple reference captions (paraphrases). Unlike some methods that necessitate paraphrases for learning descriptive captioning, SMILE naturally integrates such capability from the training corpus.
>
> **A3 (b):** We totally agree with you that the descriptiveness gains might partly come from the paraphrases. However, our work didn't utilize these paraphrases but treated them as distinct training samples, hence we didn't mention them specifically. On this topic, we tested with **a no-paraphrase corpus**, keeping just one caption per image for training in MSCOCO. Subsequent table results confirm that the model could still boost descriptiveness without paraphrases.
>
> Method|Paraphrase|Train. Data|Cap. Len.|Lex. Div.|R@1|R@5|CLIPScore|PPL
> -|:-:|:-:|:-:|:-:|:-:|:-:|:-:|:-:
> BLIP|√|567K|10.0|1.4|6.7|16.6|77.2|95.8
> BLIP-SMILE|√|567K|22.3|4.5|10.0|24.5|75.0|95.6
> BLIP-SMILE|×|113K|17.9|3.3|8.9|21.3|75.2|90.3
>
> > **Q4:** In video captioning, human perplexity is significantly higher.
>
> **A4:** Since it's common for human language perplexity to be higher than that of language models, especially in video descriptions where more intricate semantics are conveyed (each MSR-VTT video is annotated with 20 captions, indicating great diversity), we believe such relatively high perplexity is reasonable. While perplexity isn't a high-precision metric, we use it to demonstrate that SMILE captions don't have serious fluency issues, rather than comparing different models.
>
> Once again, we sincerely appreciate your meticulous feedback. Please let us know if you have further inquiries or remarks.

---

### Official Review · Reviewer_XLEL · 2023-07-04

**Soundness:** 2 fair
**Presentation:** 3 good
**Contribution:** 3 good
**Rating:** 5
**Confidence:** 4

**Summary:**

This paper proposes a novel further training approach called SMILE (Submodular Mixup Language Evaluation) for descriptive image captioning, which aims to improve the diversity and quality of generated captions. SMILE introduce to suppress conciseness optimization while encourge richness optimization to incentivize the model to generate more descriptive captions. It is based on submodular optimization and mixup techniques. It is worth mentioning that，the proposed method is architecture-agnostic and can be applied to any captioning model with MLE optimization. Experiments, conducted on two different datasets, show that the proposed approach can achieve competitive performance on three aspects: descriptiveness, accuracy, and fluency.

**Strengths:**

1. The perspective of model optimization about the unbefitting strict supervision with MLE for image captioning is intuitive. It also takes consideration into two optimizations, richness optimization and conciseness optimization.
2. A flexible and lightweight modification of MLE with a reconducted subset has been developed for further training. This modification is designed to be architecture-agnostic, meaning it can be utilized with any visual captioning model that incorporates MLE optimization.


**Weaknesses:**

1. If I understand the paper correctly, the SMILE loss suppresses the conciseness optimization and maintains the richness optimization.  The former removes the forcing optimization for descriptive and correct words that are typically not in the subset, while the latter takes into account the concise and correct words that are included in the ground-truth caption and thus in the subset. The criterion used to assess richness and conciseness appears to be relatively simple.
2. The proposed approach in this paper involves a scheme for further training existing models for image captioning. Nevertheless, several large models, including MiniGPT-4 and LLaVA, have been developed to generate more descriptive captions. Therefore, it is essential to compare the proposed approach with these existing models in the experiment.
3. During inference, if the target sequence is unknown and the subset is unavailable, does the model output a probability distribution over the whole vocabulary? This can potentially create inconsistencies between the training and testing phases.
4. In Section 3.4, the authors discussed the "semi-permeability" within the subset selection strategy and presented their findings in Figure 4. However, the meaning of "semi-permeability" is not clearly defined. Specifically, it is unclear whether "semi-permeability" refers to the ability to select and optimize models between the two extremes of simplicity and richness. Furthermore, while Figure 4 shows the relationship between the subset selection strategy and caption length, it does not shed light on the relationship between "semi-permeability" and "caption length."
5. The main text and supplementary material do not provide any qualitative results.


**Questions:**

See Weaknesses.

**Limitations:**

Yes, the authors have adequately addressed the limitations and, if applicable, potential negative societal impact of their work.

---

> ### Author Rebuttal · Authors · 2023-08-09
>
> Thank you very much for the thoughtful comments and valuable feedback. We address the specific concerns as follows:
>
> > **Q1:** The criterion used to assess richness and conciseness appears to be relatively simple.
>
> **A1:** We understand your concern, but we have to clarify that what we refer to as richness and conciseness are **relative concepts**. For example, conciseness optimization is attributed to the **MORE descriptive** prediction than the ground truth, and these "more descriptive" words are typically outside the ground truth caption. From another perspective, since we would like the captions to convey **additional details** beyond the ground truths, we need the **new words outside the sentence**. The same logic applies to richness optimization.
>
> > **Q2:** It is essential to compare the proposed approach with existing multimodal LLMs.
>
> **A2:** Thank you for the suggestion. While multimodal LLMs employ **additional training data** (e.g., detailed description) for supervised fine-tuning to align the model with the capability to describe in detail, our exploration focuses on leveraging existing data (e.g., short captions in MSCOCO) to learn more descriptive captioning. Given that our method **does not rely on extra training data**, and the model size (~227M) and pretraining data scale are **significantly smaller** than multimodal LLMs, a direct comparison might be somewhat unfair. However, we believe such a comparison remains interesting and crucial to understanding and contrasting the performance of models under different paradigms. Therefore, we've added evaluations for two popular multimodal LLMs, MiniGPT-4 and LLaVA, on the MSCOCO test set. As shown in the table, BLIP with SMILE achieves comparable self-retrieval performance with LLMs, while underperforms them obviously in both caption length (MiniGPT-4) and accuracy (LLaVA). However, despite the strong capabilities demonstrated by LLMs, the heavy computational resources they rely on are worth noting.
>
> Method|Cap. Len.|Lex. Div.|R@1|R@5|CLIPScore|PPL
> -|:-:|:-:|:-:|:-:|:-:|:-:
> BLIP (SMILE)|22.3|4.5|10.0|24.5|75.0|95.6
> BLIP (MLE+SMILE)|19.8|3.6|**10.9**|**25.1**|76.2|79.4
> MiniGPT-4-7B|**66.3**|**6.4**|10.5|23.8|75.6|14.8
> LLaVA-Lightning-MPT-7B|32.5|3.8|9.7|23.4|**79.7**|20.0
>
> > **Q3:** During inference, if the target sequence is unknown and the subset is unavailable, does the model output a probability distribution over the whole vocabulary? This can potentially create inconsistencies between the training and testing phases.
>
> **A3:** As stated in the *Method* (line 93), during inference, the model predicts across the entire vocabulary to determine the current generation. In fact, **during both training and inference, the model calculates logits over the whole vocabulary**. During training, SMILE selects part of the logits to calculate a subset probability distribution (using Softmax) for **loss calculation**, which is a post-processing of the model output and is independent of the predicting process. In other words, the **prediction behavior** of the model remains consistent during training and inference; the subset selection is only for the **evaluation** of the model's prediction (i.e., loss calculation). Hence, we believe SMILE does not introduce an inconsistency between training and inference.
>
> > **Q4: (a)** The meaning of "semi-permeability" is not clearly defined. **(b)** While Figure 4 shows the relationship between the subset selection strategy and caption length, it does not shed light on the relationship between "semi-permeability" and "caption length."
>
> **A4 (a):** As we mentioned in the *Introduction* (line 62), "semi-permeability" refers to the nature of SMILE that **allows richness optimization but blocks conciseness optimization**, similar to a semipermeable membrane. We will describe the meaning more clearly in the revised version.
>
> **A4 (b):** By designing different subset selection strategies, different **types** of "semi-permeability" are achieved, which in turn affects the caption length. Specifically, *SMILE subsetting* creates a type of "semi-permeability" that permits richness optimization while blocking conciseness optimization, leading to gradually increasing caption lengths. In contrast, *reverse subsetting* establishes an opposite type of "semi-permeability" that only permits conciseness optimization, resulting in decreasing caption lengths. Here, caption length variations serve as indicators of the presence of "semi-permeability". We will enhance this explanation in the revised version. Moreover, our *Supplementary* provides more analysis about how the **degree** of "semi-permeability" affects caption length. We present experiments that combine SMILE with MLE, illustrating the significant correlation between caption length and the proportion of SMILE loss.
>
> > **Q5:** The main text and supplementary material do not provide any qualitative results.
>
> **A5:** We've included some qualitative results in Fig. 1. To show more results of SMILE's effects, we've also provided a PDF file with additional qualitative results in ***Global Response***. These results are from our randomly selected human evaluation set, accompanied by their obtained scores.
>
> Once again, we sincerely appreciate your concerns and suggestions and hope our responses address them. Please let us know if there's more feedback.

---

> > ### Comment · Reviewer_XLEL · 2023-08-17
> > **Borderline accept.**
> >
> > Thanks for the author's reply. The author provided detailed explanations, but the definitions of conciseness and richness at the methodological level still seem somewhat rudimentary. Therefore, I tend to still keep my ratings.

---

> > > ### Author Response · Authors · 2023-08-17
> > > **Response to Reviewer**
> > >
> > > We are thankful for the reviewer's feedback.
> > >
> > > We would appreciate it if the reviewer could elaborate more on the comment "definitions of conciseness and richness at the methodological level still seem somewhat rudimentary".
> > >
> > > For better understanding, let’s walk through the logic of our method again, taking conciseness as an example:
> > >
> > > 1. We have explicitly defined our concept of conciseness optimization (lines 5, 40): if the model predicts a word (e.g., "white") that is more descriptive than the label (e.g., "cake"), it will be penalized and optimized to prefer concise expressions.
> > > 2. To prevent conciseness optimization, we need to exclude words that could lead to such optimization, i.e., words that are "more descriptive" than the ground truth.
> > > 3. Then, which words are "more descriptive"? Words deemed "more descriptive" than the ground truth typically convey additional details that are absent in the ground truth caption. Therefore, naturally, they are new words outside the ground truth caption. (Please note that while we claim that "more descriptive" words are typically outside the ground truth caption, we are **not** implying that all new words outside the caption represent "more descriptive" words. We wonder if the reviewer's comment on our method as "rudimentary" might stem from this potential misunderstanding?)
> > > 4. Hence, we select a subset only containing words in the ground truth sentence, effectively excluding the "more descriptive" words and subsequently preventing conciseness optimization.
> > >
> > > We consider that such a subsetting strategy is a neat solution for controlling conciseness and richness optimization. We therefore respectfully disagree to view it as "rudimentary" and believe, as echoed by other reviewers that it is "very subtle", "interesting exploration", and "idea is effective".
> > >
> > > We hope the above explanation addresses the reviewer's concerns. Or if the reviewer can elaborate on the concerns more specifically, e.g., by pointing out certain scenarios where our method might be flawed, we would greatly appreciate it and believe we can provide a more focused explanation.
> > >
> > > Additionally, in our revision, we have made thorough improvements to the related descriptions for better clarity. In the *Method* section, we've also defined richness and conciseness optimization in a more precise mathematical format. We hope that the revised version will further prevent any potential misunderstandings.
> > >
> > > We kindly request the reviewer to carefully assess whether we have addressed your concern and please let us know if there are any additional questions.
> > >
> > > Thank you！

---

> > > > ### Comment · Reviewer_XLEL · 2023-08-18
> > > > **Official Comment by Reviewer XLEL**
> > > >
> > > > Thanks for your reply, my concerns are addressed. But I suggest that the definitions of conciseness and richness can be summarized from the perspective of captioning tasks, rather than defined by examples and hypotheses. Authors can try to revise these in the final version.

---

> > > > > ### Author Response · Authors · 2023-08-18
> > > > > **Response to Reviewer**
> > > > >
> > > > > We thank the reviewer for further feedback.
> > > > >
> > > > > We would like to argue that our definitions of conciseness and richness concept are framed from the perspective of captioning tasks; the "hypotheses" we introduced serve as the causes behind these concepts; the "examples" we provided aim to facilitate easier understanding.
> > > > >
> > > > > Let us walk through the relevant details:
> > > > >
> > > > > 1. As stated in the paper, the direct definition of conciseness optimization is the model "be penalized and optimized to prefer concise expressions" (line 6) or optimizing the model "towards a more concise captioning behavior" (line 41).
> > > > > 2. We introduce two types of scenarios (referred as "hypotheses" by the reviewer) as the causes of conciseness and richness optimization (e.g., "if the model predicts a word ..." at line 40). This also echoes and aligns directly with our designed methodology.
> > > > > 3. The examples provided were meant to aid readers to better understand the nuanced concept. We believe the additional inclusion of examples in a definition is a good way to make complex or unfamiliar concepts more comprehensible.
> > > > >
> > > > > We appreciate the reviewer's suggestion on clarifying the definition. We will make corresponding revision in the updated version.

---

### Official Review · Reviewer_5fp8 · 2023-07-06

**Soundness:** 2 fair
**Presentation:** 2 fair
**Contribution:** 2 fair
**Rating:** 5
**Confidence:** 3

**Summary:**

The authors proposed the SMILE optimization method to enrich the detailed description of images in Captioning models. The authors analyzed the optimization process in the Image Captioning task and divided the optimization process into conciseness optimization and richness optimization. By suppressing the conciseness optimization process, the authors help the model to enrich the detailed description of the text. Specifically, the authors further train the already trained captioning model and then only calculate the loss of vocabulary on GT during the optimization process, thus avoiding the simple GT suppressing the model from outputting detailed vocabulary.

**Strengths:**

1.	The authors proposed the SMILE optimization method, which can significantly improve the model's degree of image description and enrich the vocabulary richness of the output Caption.

2.	The ablation study built by the authors provides a detailed discussion of the working principle of the SMILE method. By selecting the right subset selection strategy, the model's output caption's degree of image description can be controlled. Furthermore, the effectiveness of this method in other tasks (Video Captioning) is demonstrated.

3.	The SMILE method is a fine-tuning method, so it can help a model to converge quickly and enrich the description of images in Captioning models.


**Weaknesses:**

1.	SMILE shows a significant decline in the CLIPScore metric across multiple experiments, and there is also a significant drop in the Acc metric for semantic evaluation in Table 2, indicating that SMILE has some issues concerning semantic correctness.

2.	The authors only tested the effect of SMILE on the BLIP model. It would be better to test a few more models to ensure the robustness of this method.

3.	In the "Models 'absorb' details from the corpus" section, a more refined metric might be needed to highlight the effect. Using only one visualization example lacks persuasiveness.


**Questions:**

1.	Can it be demonstrated that SMILE does not have a significant loss in semantic correctness?
2.	Is there a more specific metric to show why SMILE can help enrich the image description?





**Limitations:**

1.	From the current experimental results, the SMILE method has a significant loss in the correctness of image descriptions. Therefore, how to help the model enrich the image description while ens

---

> ### Author Rebuttal · Authors · 2023-08-09
>
> Thank you very much for the thorough understanding and insightful comments. We address the specific concerns and questions as follows:
>
> > **Q1: (a)** SMILE shows a significant decline in the CLIPScore metric across multiple experiments, and there is also a significant drop in the Acc metric for semantic evaluation in Table 2, indicating that SMILE has some issues concerning semantic correctness. **(b)** Can it be demonstrated that SMILE does not have a significant loss in semantic correctness?
>
> **A1 (a):** As highlighted in our paper, "**Talks much errs much.**" The MLE-optimized model tends to output generic captions of about 10 words (e.g., "a woman holding a cake with lit candles"), which usually contain simple concepts to describe the main features in images, and thus are less prone to errors. In contrast, SMILE captions, with an average length exceeding 20 words, contain rich additional details, amplifying the risk of errors, especially when these details are inherently harder to identify (e.g., small objects) and express (e.g., less frequent words).
>
> Therefore, we believe the observed decline in CLIPScore (-2.2 on MSCOCO and -0.6 on Flickr30K) is reasonable and acceptable. Also, it's important to note that since the CLIP model isn't impeccable at comprehending all details in images and captions (especially richer SMILE captions with more low-frequency words), we believe that the reliability of CLIPScore needs to be viewed with caution.
>
> In human evaluations, we noticed that the annotators often base accuracy assessment on **error spotting**, meaning simple captions easily get perfect scores (i.e., 5). However, for very detailed captions, even a slight inaccuracy can reduce scores (to 4 or less), greatly disadvantaging SMILE in human evaluations. To illustrate this better, we provide some real examples from our human evaluations in ***Global Response***.
>
> **A1 (b):**
> - Firstly, SMILE has shown significant improvement in **self-retrieval performance**, suggesting that the generated captions are fundamentally correct and more discriminative for retrieving images. An imprecise caption would struggle to retrieve the original image, especially at Recall@1.
> - Secondly, in human evaluations, an accuracy score beyond **4 (*good*)** demonstrates the good quality of SMILE captions, even though they confront the risk of "talks much errs much".
> - Lastly, our experiments in *Supplementary* show that a **compromise** between descriptiveness and accuracy can be achieved by simply combining SMILE with MLE, and such a compromise can lead to **enhanced accuracy and self-retrieval performance**.
>
> Beyond the above responses, we'd also like to mention that while SMILE helps models in altering their **generation habits** (i.e., producing longer and more detailed captions), the quality of the caption still relies a lot on the basic model's **inherent capability**. We view SMILE as a preliminary exploration and leave further accuracy enhancements to our future endeavors.
>
> > **Q2:** The authors only tested the effect of SMILE on the BLIP model. It would be better to test a few more models to ensure the robustness of this method.
>
> **A2:** Thank you for pointing this out. In this paper, we validated only on BLIP-base and its video task variant. However, ensuring robustness across different models and model scales is essential. As a result, we applied SMILE to **BLIP-large** and another popular VLP model, **OFA**, on MSCOCO. As shown in the table, SMILE consistently improves the generation length, lexical diversity, and descriptiveness of the models. In the revised manuscript, we will include a new section in the appendix with more models and analyses.
>
> Method|Cap. Len.|Lex. Div.|R@1|R@5|CLIPScore|PPL
> -|:-:|:-:|:-:|:-:|:-:|:-:
> BLIP-large|10.0|1.4|6.7|17.3|**77.4**|92.3
> BLIP-large-SMILE|**24.0**|**4.9**|**9.2**|**22.4**|74.7|100.0
> OFA-base|10.1|1.2|5.9|15.6|**76.7**|64.6
> OFA-base-SMILE|**16.5**|**4.0**|**6.9**|**16.2**|71.6|109.7
> OFA-large|9.8|1.2|6.4|16.0|**77.1**|68.6
> OFA-large-SMILE|**18.8**|**5.9**|**9.3**|**21.9**|73.6|125.9
>
> > **Q3:** In the "Models 'absorb' details from the corpus" section, a more refined metric might be needed to highlight the effect. Using only one visualization example lacks persuasiveness.
>
> **A3:** As introduced in the preamble of Section 3.4 *Further Analysis*, this section digs into **where the model learns the usage of detailed words**. To answer the question, we conducted experiments to explore how the presence of detail in the corpus impacts the efficacy of SMILE. We constructed two datasets for this purpose: Simplest-COCO, which contains no details, and Simpler-COCO, with only a few details. The results demonstrate that on Simplest-COCO, SMILE does not induce the model to produce longer outputs. However, even with minimal details present in the corpus (Simpler-COCO), SMILE is able to incentivize the model to express more details. This suggests that the model's capability to express details **stems from the training corpus**. If we understand your inquiry correctly, the metric we employ to highlight the effect is the caption length variation, i.e., as the caption becomes progressively longer, it indicates the model is "absorbing" the capacity to generate details. We will refine this expression for better clarity in the revised version. Lastly, as for our visualization, it's primarily meant to showcase **how SMILE exerts its effect from token-level loss**, helping better understand our methodology; it should not serve as the visualization of the "absorb" section.
>
> Once again, we sincerely appreciate your concerns and suggestions and hope our responses address them. Please let us know if there's more feedback.

---

> > ### Comment · Reviewer_5fp8 · 2023-08-17
> >
> > Most of my concerns are solved and thus I decide to rasie the final rating to boardline accept.

---

> > > ### Author Response · Authors · 2023-08-18
> > > **Response to Reviewer**
> > >
> > > Thank you for your comments. We appreciate that you consider raising your rating.

---

### Official Review · Reviewer_cENC · 2023-07-10

**Soundness:** 3 good
**Presentation:** 4 excellent
**Contribution:** 3 good
**Rating:** 6
**Confidence:** 4

**Summary:**

This paper argues that captioning models with maximum likelihood estimation as the training objective led to conciseness optimization, which limits the model to generate descriptive captions. To this end, they introduce semipermeable maximum likelihood estimation (SMILE), which allows richness optimization for generating longer and more detailed sentences.  As a result, they conduct extensive experiments on MSCOCO and Flickr30K datasets, which demonstrate the SMILE significantly enhances the descriptiveness of generated captions.

**Strengths:**

1.	**The writing is clear and well-motivated**. It is urgent to explore a new training object function for image captioning optimization. And conciseness optimization is a core problem that needs to be addressed in image/video captioning.
2.	**The proposal SMILE is very subtle**. SMILE considers the probability over a vocabulary subset with limited words (words in the ground truth caption), which is an interesting exploration. The block-level conciseness optimization idea is effective.
3.    **The result is convinced**. The result demonstrates that SMILE can generate significantly longer and more descriptive captions.
4.   **The SMILE is architecture-agnostic and plug-and-play**.  It can generalize to many existing VLP models.

**Weaknesses:**

1. **The SMILE is not generalized to the large language models**. It doesn’t observe significant improvement in response length or overall quality.
2. **Generalization of the video captioning.** Although they conduct experiments on MSR-VTT for video captioning, it will be interesting to extend SMILE to dense video captioning (e.g. Activity-Net).

**Questions:**

As shown in Weakness.

---

> ### Author Rebuttal · Authors · 2023-08-09
>
> Thank you very much for the thoughtful comments and valuable suggestions. Our responses to the specific questions are as follows:
>
> > **Q1: The SMILE is not generalized to the large language models.** It doesn’t observe significant improvement in response length or overall quality.
>
> **A1:** The initial motivation for this paper was based on the idea that MLE is not perfectly suitable for **image captioning**, as "a picture is worth a thousand words". Hence, we introduced SMILE to encourage models to generate longer captions with more details. In addition, we have conducted **preliminary verification** on the Supervised Fine-Tuning (SFT) of Large Language Models (LLMs) as **an intriguing exploration in the general question-answering domain**. We have demonstrated in *Limitation* and *Supplementary* that SMILE may not be directly generalizable to other tasks, and we analyzed the possible reasons, hoping to offer some empirical experiences for future research. However, it's essential to note that, as a method specifically designed for the captioning task, the generalizability of SMILE for broader tasks is beyond the scope of this paper. We leave it to our future studies.
>
> > **Q2: Generalization of the video captioning.** Although they conduct experiments on MSR-VTT for video captioning, it will be interesting to extend SMILE to dense video captioning (e.g. Activity-Net).
>
> **A2:** Dense video captioning requires the model to describe multiple events from an untrimmed video, where a more detailed description would be beneficial. Thus, extending SMILE to this domain is indeed exciting. However, being a combination of two sub-tasks, namely **event detection** and **event captioning**, the mainstream solutions for dense video captioning either adopt a two-stage approach, where the second stage captioning **boils down to a typical video captioning task**, or they jointly address both sub-tasks, resulting in **the language generation being influenced by inter-subtask interaction**. For example, the state-of-the-art method Vid2Seq [1] augments a language model with special time tokens, incorporating event localization into sequence generation. Therefore, applying SMILE to dense video captioning is not a straightforward and simple task and requires specific and meticulous design for different models. Additionally, given the length and rich semantics of dense video captions, evaluating the generated results presents its challenges.
>
> We appreciate this valuable suggestion. We will explore the performance of SMILE in dense video captioning in future work.
>
> [1] Vid2Seq: Large-Scale Pretraining of a Visual Language Model for Dense Video Captioning, CVPR 2023.
>
> Once again, we sincerely appreciate your comments and suggestions. Please let us know if there's more feedback.

---

### Author Rebuttal · Authors · 2023-08-09

We sincerely thank all the reviewers for their valuable feedback, which is crucial for further refining our work. After the initial submission, we dedicated considerable time to improving the paper's structure and writing. Consequently, some of the potentially confusing and ambiguous areas highlighted in the reviews should now be enhanced. Moving forward, we will thoroughly consider the reviewers' comments and suggestions to make further revisions to our manuscript.

Furthermore, to better address the reviewers' questions, we provide supplementary materials as an attachment, including some qualitative results from our human evaluations. We hope these results will assist the reviewers in gaining a deeper understanding of our work.

---

### Decision · Program_Chairs · 2023-09-21

**Decision:**

Accept (poster)

**Comment:**

The paper presents a novel training technique called SMILE, which is specifically designed for captioning. By blocking conciseness optimization of MLE through subsetting vocabulary, SMILE effectively allows the final trained model to generate more descriptive captions than the ground-truth captions themselves. The paper also convincingly shows the effectiveness of the method and valid analyses.

As Reviewers cENC and nk4n mentioned in their reviews, conciseness optimization is a core problem that needs to be addressed in image/video captioning. Also as pointed out by Reviewer 5fp8, the ablation study and analyses provide a sound discussion of the working principle in the SMILE optimization, and demonstrate its effectiveness.

Overall, the work is timely and will definitely draw attention from the community. Based on the positive consensus made across all the reviewers, the AC also recommends the acceptance of the paper.